# Can We Reliably Detect Respiratory Diseases through Precision Farming? A Systematic Review

**DOI:** 10.3390/ani13071273

**Published:** 2023-04-06

**Authors:** Luís F. C. Garrido, Sabrina T. M. Sato, Leandro B. Costa, Ruan R. Daros

**Affiliations:** Graduate Program in Animal Science, School of Medicine and Life Sciences, Pontifícia Universidade Católica do Paraná, Curitiba 80215-901, Brazil; luis.garrido@pucpr.edu.br (L.F.C.G.); sabrina.sato@pucpr.edu.br (S.T.M.S.); batista.leandro@pucpr.br (L.B.C.)

**Keywords:** sensor, smart farming, bioacoustic, animal welfare, health monitoring

## Abstract

**Simple Summary:**

In this systematic review, we assessed studies on automatic monitoring of respiratory disease in livestock. This can help to understand if precision livestock farming (PLF) technologies are able to fulfill their purpose and can provide insights into the potential of commercially available PLF options. Few PLF technologies had good performance measures with tests under field conditions and using a reliable reference test, indicating that technologies still fall short for monitoring clinical signs or the onset of respiratory diseases in bovine, swine, and poultry productions. This review assessed previously published development and validation articles of PLF technologies to monitor respiratory health, and it highlights that more than just performance measures should be assessed when discussing the potential and pitfalls of PLF technologies.

**Abstract:**

Respiratory diseases commonly affect livestock species, negatively impacting animal’s productivity and welfare. The use of precision livestock farming (PLF) applied in respiratory disease detection has been developed for several species. The aim of this systematic review was to evaluate if PLF technologies can reliably monitor clinical signs or detect cases of respiratory diseases. A technology was considered reliable if high performance was achieved (sensitivity > 90% and specificity or precision > 90%) under field conditions and using a reliable reference test. Risk of bias was assessed, and only technologies tested in studies with low risk of bias were considered reliable. From 23 studies included—swine (13), poultry (6), and bovine (4) —only three complied with our reliability criteria; however, two of these were considered to have a high risk of bias. Thus, only one swine technology fully fit our criteria. Future studies should include field tests and use previously validated reference tests to assess technology’s performance. In conclusion, relying completely on PLF for monitoring respiratory diseases is still a challenge, though several technologies are promising, having high performance in field tests.

## 1. Introduction

Worldwide livestock respiratory diseases are highly prevalent, reducing productivity and increasing death risk [1]. In poultry production, animal-level prevalence ranges from 3% [2] to 49.3% [3] while flock-level respiratory diseases prevalence often reaches over 80% [4]. In swine, 38.5% [5] of animals are affected by this malady, and in cattle the prevalence ranges from 0.5% to 61% [1]. Mortality due to respiratory diseases is relatively high in livestock, reaching 47% in poultry [6], 4% in swine [5], and 19% in bovine [7,8].

Many pathogens are responsible for causing respiratory diseases in livestock [9,10,11], but the clinical manifestation is similar in most cases. The main clinical signs are coughing, sneezing, lethargy, and nasal and ocular discharge [9,11].

Precision Livestock Farming (PLF) is defined as automated continuous animal monitoring to aid with farm management [12]. PLF technologies are based on the use of sensors to monitor animals and have been developed for monitoring respiratory health in dairy calves [13], poultry [14], and swine [15]. These technologies can monitor respiratory diseases by monitoring signs, such as cough, sneezing, and fever or detecting specific types of diseases such as Newcastle disease and porcine wasting diseases. These technologies can detect the presence of diseases or clinical signs at the group level [15,16] or individual level [16].

To assess if a technology has external validity (i.e., is effective to be implemented in commercial farms), several points should be observed: high performance measures and the validation conditions must be similar to the conditions in commercial farms [17].

Performance measures include specificity (Sp), sensitivity (Se), precision, accuracy, positive predictive value (PPV), and negative predictive value (NPV). According to Dominiak and Kristensen [18], if the performance measures are not satisfactory there is no need to further assess the time window and validation conditions. The performance measurements ultimately arise from the results of a reference test (usually called a gold standard), which is used as a reference to assess if the developed technology was efficient in achieving its objectives [19]. Thus, when evaluating PLF development, it is critical to assess the adequacy of the chosen reference test.

The study condition may determine if the performance is replicable in a real-life situation (i.e., commercial setting); therefore, the efficiency from technologies validated with field condition tests is likely to be similar to the efficiency on commercial farms [18].

Performance measures may also be influenced by the type of data collected and the type of sensors applied. For example, sensors that are only capable of recording data related to nonspecific signs of disease, such as elevated body temperature, are not indicated for detecting a specific type of disease [20]. In addition, performance measures that consider the total number of cases (i.e., error rate, number of false positives for every true positive prediction) are biased by the true disease prevalence, with rarer diseases having higher error rates compared to diseases that are highly prevalent [18]. Therefore, critically combining all this information with performance measures allows one to assess technology readiness.

There have been recent systematic reviews about PLF on disease monitoring [20,21,22,23]. However, these reviews focused on technologies that target many welfare-related issues at once (e.g., lameness, respiratory diseases, diarrhea). A more profound investigation of PLF devices developed exclusively for monitoring respiratory health status is needed to enhance the availability of information on the efficiency of these technologies. Understanding the effectiveness of a technology is much more complex than only looking at its performance. To support that the technology is reliable and able to perform well in real-life conditions, the study should present the study’s condition, reference test, and a full description of how the technology was tested.

The objectives of this systematic review were to assess: (1) the advances of PLF for monitoring respiratory diseases at group or animal level in different livestock productions, (2) the quality of studies in reporting the effectiveness of the technology, and (3) if it is possible to rely on PLF to automatically monitor respiratory diseases in livestock species.

## 2. Materials and Methods

### 2.1. Literature Search

A systematic literature review was conducted to gather data on validated PLF technologies for the detection of respiratory diseases in livestock species.

The PRISMA guideline [24] was used for the systematic literature search. The search was conducted in the Scopus, Web of Science, and IEEE databases on 21 September 2022. The search field in Web of Science was “topic”, and in Scopus was “article title, abstract, keywords”. Search terms are presented in Table 1.

These terms were sorted as “species” terms, “technology” terms and “type of condition” terms. The search string consisted of a combination of terms separated by the Boolean operator “AND” and “OR”; for example: “Dairy Cow*” OR “Cow” AND “Precision Livestock Farming” OR “Non-invasive Technology” AND “Respiratory Disease*” OR “Cough”. To filter the search, in Web of Science and Scopus platform the option “Articles” and “Conference paper” was selected in “Document Types”.

The inclusion criteria were original research studies published at any time about the development or validation of PLF technologies for monitoring respiratory disease in production animals (this included technologies aimed at monitoring clinical signs and technologies developed for disease detection). The exclusion criteria were studies that did not develop, validate, or apply PLF technologies to exclusively detect respiratory diseases or respiratory diseases clinical signs (i.e., research papers on general health monitoring or other disease were not included); this included research articles about the development or improvement of a specific technology’s feature, for example, localization of the cough event or differentiation between types of coughs were excluded. Studies of technologies developed for other species that are not present in the “species” terms, were also excluded.

Two researchers were involved in the initial search and screening process (L.F.C. Garrido and S.T.M. Sato). The researchers worked independently on the search using the same terms and databases. The inclusion and exclusion of articles were also conducted independently by each researcher. The results of the screening process were compared after the search and inclusion or exclusion of articles. This step was conducted to ensure that all relevant studies were included. The screening started with the exclusion of duplicate studies. Secondly, studies that were not related to PLF were excluded by title and abstract. Then studies were screened following the inclusion and exclusion criteria after analyzing the title, abstract, and keywords [25]. The remaining studies were included or excluded after analyzing the entire content of the article. Conflicted studies were analyzed together by screeners and were included or excluded from the review after agreement if the study complied with the inclusion or exclusion criteria. Information about the number of excluded and included studies and the reason for excluding studies is presented in Figure 1.

Three articles were excluded because we were unable to find the full text. One article was just available in Mandarin and therefore excluded to prevent mistakes due to inaccurate translation.

All articles included in the review were used for assessing the advances of PLF technologies to monitor respiratory diseases.

### 2.2. Data Gathered from Articles

After the screening process, the following data were extracted from included articles: species the technology was applied in, conditions in which the study was conducted (laboratory or field condition: a study was considered performed in laboratory conditions when performed in a university or research center or conducted in a controlled environment not similar to a commercial farm; field conditions were considered when the study was conducted in a commercial farm, or field conditions were simulated in a controlled environment), the type of sensor/device used (sound-based, image-based, or any other sensor found in the literature), performance measures (Se, Sp, precision, accuracy, PPV, PNV, or any performance measure presented by the study, if recall was presented we considered it as Se), and the reference test used for validating the technology.

The animal production stage in which these technologies are tested was also assessed. For swine production, we determined the production stage based on animals’ age or weight when the articles did not clarify the stage. Animals were considered in the fattening stage when animals were ten to sixteen weeks old and weighed between 30 and 60 kg. Animals with age above sixteen weeks and that weighed more than 60 kg were considered in the finishing stage. For bovines, the stage was considered the type of animal used (e.g., calves, steers) and poultry the age.

Information regarding the production system and the aim of the production (e.g., egg, broiler, dairy, or meat production) was also gathered when presented by the article. For studies conducted in laboratory conditions, the production system was considered as an experimental setup.

### 2.3. Risk of Bias

We assessed the risk of bias in studies included in this review. This process was held to understand the consistency of the results based on the study design [26]. A study is considered to have a low risk of bias when all information that could bias the results is presented. Similar to Stygar et al. [23] and Hendriks et al. [27], we considered that the article had to provide the following information to prevent bias: conditions in which the study was held (laboratory or field conditions), housing, type of sensor, and how it was installed (manufacturer of the sensor used, how many meters from the ground, where was it installed), the software used (what type of software, what machine learning method was used). In addition, we added three more pieces of information that had to be presented to assess bias; namely, the population description (age, weight or production stage), number of animals and raw data used to calculate performance measures. A study was categorized to have a high risk of bias if one or more essential information was not presented.

### 2.4. Technology Reliability

In this review, the word “reliable” refers to the sole use of PLF technologies to detect respiratory diseases or to monitor the clinical signs of disease. Performance, study conditions, and the reference test used were assessed to decide if a technology was considered reliable. Only studies with a low risk of bias were assessed for reliability.

For dairy cows, the International Organization for Standardization (ISO 20966, 2007) [28] presents a minimum performance required to support a technology efficiency (Se > 80% and Sp > 99%). These minimum requirements are supported because the reported success that farmers have identifying clinical mastitis is 80%, and therefore a technology with a performance above this threshold is able to outperform the human [18]. To assess performance of PLF technologies for monitoring respiratory diseases, it would be ideal to use a similar rationale to that of the ISO 20966, though to our knowledge there are no specific standards for monitoring respiratory diseases in livestock. Therefore, similarly to Stygar et al. [23], we assessed the validated technology’s performance by considering high performance when Se > 90% and Sp or precision was >90%. Thus, both Se and Sp had to be higher than 90% for the technology presented. If Sp was not presented but the article presented precision, both Se and precision had to be higher than 90%. Articles that did not present Se and Sp or precision were not included in the discussion of reliable technologies. When a study compared various methods for developing the technology (e.g., when multiple machine learning methods were used), we collected the performance of the technology that achieved the highest performance.

The reference test was considered reliable when the study presents evidence that it is effective to assess the clinical sign that is being monitored. For an article to prove reference test reliability, tests should be presented demonstrating that the used reference test was able to detect the disease or clinical sign. Another way of considering reliability was the use of a well-established reference test, given that the authors provided peer-reviewed references for it. For sound-based technologies, remote audio labeling sounds from audio recordings were not considered reliable due to misclassifications that may occur in the labeling process [29].

In sum, the criteria for considering a technology reliable were high performance, use of a reliable reference test, tests conducted in field conditions, and the study having low risk of bias.

## 3. Results

A total of 23 articles were included in this review. Most of the articles were on PLF technologies for swine (13/23), followed by poultry (6/23) and bovine production (4/23). No articles were found about PLF technologies for small ruminants (goats and sheep).

Most technologies used sound-based devices (21/23). Other types of sensors found were image-based (1/23) and accelerometer (1/23). Table 2 summarizes the data gathered from the studies included in this review.

The production stage, production system, and aim of the production of all species are presented in the Appendix A available at [https://doi.org/10.6084/m9.figshare.21758543.v2, accessed on 14 March 2023].

### 3.1. Studies’ Condition and Reference Tests

In this section, we report all reference tests found in studies included in this review. The reference test varied in relation to the species and the technology applied. Reference tests found were: blood analysis (1/23; used the number of neutrophils), clinical assessment (2/23; assessment of clinical signs presented by the animals), clinical assessment and blood analysis (1/23; number of neutrophils and clinically assessment for respiratory diseases), remote audio labeling (11/23; manually label sounds from audio files), live audio labeling (3/23), PCR (2/23), remote audio labeling and blood analysis (1/23; label sounds and collected blood samples to identify the disease), video labeling and blood analysis (1/23; used both image of the animals and audio recordings to label cough sounds and collected blood samples to identify the disease), video labeling, and PCR (1/23).

### 3.2. Risk of Bias

A total of eight studies were considered to have a high risk of bias: five swine production studies [32,33,37,38,41], two poultry production studies [43,44], and one for bovine production study [47]. Table 3 shows the specific information that was presented or not presented in each study.

### 3.3. Respiratory Disease PLF Technologies for Poultry Production

All the technologies found for poultry production were sound-based (6/6). However, depending on the methodology, the monitored sounds could be cough and snore (1/6), vocalization (3/6), sneeze (1/6), or rale sounds (1/6). The performance measures of the technologies for poultry production are presented in Table 4.

From the studies included, Liu et al. [45] achieved good performance measures (Se, precision, accuracy, and F1-score) and was developed in field conditions.

Banakar et al. [42] and Cuan et al. [44,46] developed technologies that aimed to detect and diagnose different types of respiratory diseases in poultry. The technologies were developed based on vocalization before and after virus inoculation under laboratory conditions and used PCR tests as reference test. Cuan et al. [44] achieved a high accuracy for the detection of Avian Influenza but did not present other performance measures (Se, Sp, and precision). Banakar et al. [42] achieved a high performance (Se and Sp) for detecting Avian Influenza but failed to achieve a high Se for Bronchitis Virus and Newcastle Disease. Cuan et al. [46] achieved high performance for all performance measures presented, with a technology aimed at detecting Newcastle disease.

All other technologies were based on specific sounds for monitoring respiratory disease. Rizwan et al. [43] monitored rale sounds and achieved high accuracy (97.6%) but was unable to achieve high Se and precision. Carpentier et al. [14] developed a technology that detects sneeze sounds; however, the technology Se was low and therefore ineffective to identify true positive cases of the disease. Both studies were conducted under laboratory conditions.

### 3.4. Respiratory Disease PLF Technologies for Bovine Production

Several types of sensors were used in bovine studies, sound-based technology (2/4), image-based (1/4), and accelerometer (1/4). The performance measures of the technologies for bovine production are presented in Table 5.

Almost all technologies developed for bovine production aimed to detect respiratory diseases in dairy calves [13,16,48]. This is in line with the period of highest risk for respiratory diseases in bovines [49].

The sound-based technologies [13,48] were developed and validated in field conditions; however, both technologies were inefficient to correctly identify true positive cases of the disease.

Another study used an accelerometer embedded in a necklace to detect respiratory diseases in dairy calves [16]. The accelerometer monitors feeding and activity behavior. This study achieved a high Sp; however, like the technologies from Vandermeulen et al. [13] and Carpentier et al. [48], the Se was low.

### 3.5. Respiratory Disease PLF Technologies for Swine Production

All studies assessed were sound-based (14/13). The performance measures of the technologies for swine production are presented in Table 6. Most technologies were developed for the fattening stage (9/13), followed by finishing stage (3/13), and in one study no information was provided about the production stage.

Many technologies for swine production presented good performance measures [15,37,39,40]. Swine respiratory disease automatic detection has been researched for over two decades [30,31,32,33], enabling a myriad of studies on this topic.

Most technologies aimed at detecting coughs to monitor respiratory health and thus were not developed to detect a specific respiratory disease [30,31,32,33,34,35,36,38,39,40]. The performance mostly assessed by these studies was the cough detection rate, which refers to the accuracy of the technology in detecting cough sounds. Many studies just presented this value to “prove” technology’s efficiency; therefore, we were incapable to evaluate if the technology could be considered reliable by the standards used in this study.

The technologies developed by Shen et al. [39,41] and Yin et al. [40], achieved high performance in the detection of cough sounds. However, the studies used audio labeling recordings as the reference test, perhaps resulting in a less reliable dataset [29].

Another set of studies from the same research group [15,37] developed a technology that aimed to identify specific swine respiratory diseases (Mycoplasma Hyopneumoniae, Postweaning Multisystemic Wasting Syndrome and Porcine Reproductive and Respiratory Syndrome virus) by monitoring cough sounds. Both studies achieved an overall performance > 90% for Se, precision, and F1-score. Because their technology aimed to detect the disease, the reference test used was serological analysis of suspected pigs to detect the disease. The cough detection rate was also >90% for both studies. Different from the reference test of Shen et al. [39,41] and Yin et al. [40], the audio was labeled with audio and video recordings and therefore could lead to a more reliable dataset due to the video footage that could help conclude if an event is a cough sound.

## 4. Discussion

We found promising results regarding PLF technologies developed for monitoring respiratory diseases in livestock production. Some studies were able to achieve a high performance; however, when assessing key points of the validation process (reference test and study condition), many technologies were not considered reliable for automatic detection of livestock respiratory diseases. Nine studies did not perform the validation process in field conditions. We highlight that tests in laboratory conditions are important for an initial analysis of technology’s potential, but tests in field conditions are required to assess technology performance in real-life situations.

Most of the technologies applied for respiratory disease detection are sound-based. The most common respiratory diseases’ clinical signs lead animals to emit sounds [9,14,50], and thus it is possible to monitor with the use of microphones. Diagnosis of respiratory disease is more complex than monitoring only one clinical sign and requires assessment of multiple signs (e.g., cough, fever, nasal discharge, lethargy, lung consolidation). Two articles excluded from this review monitored coughing with PLF technologies to assess air quality in swine productions [51,52]. Albeit air quality is a major risk factor for respiratory diseases, an alert based on air quality does not necessarily mean that the animals are sick.

Monitoring more than just one clinical sign of respiratory disease is very important to create a technology that could lead to a more reliable diagnosis; for example, monitoring cough and fever together could be useful for developing a technology able to better diagnose respiratory diseases. Other data, not collected by the technology, could also be used to overcome false positives, for example: season, feeding time (feeding may increase fine particles in the air causing irritation and thus coughing when inhaled), and animal handling. One article used a camera to monitor animals’ temperature [47]. Even though this study presented information that confirms the reliability of temperature as a clinical sign that can be monitored for detecting respiratory diseases, many other diseases might be responsible for elevated body temperature. Non-specific, clinical signs are useful for veterinarians to support decisions on the health status of an animal; however, when developing technology, monitoring specific clinical signs is preferred for detecting targeted diseases [20].

Due to the current concerns about antimicrobial resistance [53], methods to identify diseased individuals could be beneficial for reducing the overall antibiotic use on the whole group as only a single animal would be treated. None of the reviewed studies on poultry and swine attempted to identify the individual with the ailment, and only one study on dairy calves was able to detect the individual because of the type of sensor used: an accelerometer attached to the calf [16]. Current management practices make it hard for the development of PLF technologies for monitoring respiratory health that can detect sick individual using bioacoustic sensors. Perhaps a technology able to identify the individual through voice recognition could be developed for identifying the individual affected by a disease. We understand that the development of such a solution would require much research to overcome usability and technological challenges.

### 4.1. Performance Measures

Overall, Se was the most presented performance measure; this test shows the capacity that the algorithm must correctly diagnose diseased individuals [54].

Compared to Se, Sp was presented in fewer articles. Sp indicates the capacity of the technology to correctly identify individuals that are not diseased [54]. A technology with low Sp could lead to an increasing number of false alarms, which in turn may reduce system’s usage (e.g., farmer stop paying attention to the alerts) or antibiotic overuse.

In assessing the system’s validity, Se and Sp may give different but complementary insights. Evaluating these performance measures together demonstrates how reliable an alarm given by a technology is, so we hypothesize that high Se and Sp can potentially increase customer’s confidence in the alarms given.

The equation for Se is presented as “Se = true positive/true positive + false negative”. Technologies with low Se will fail to identify diseased animals or monitor clinical signs because too many false negatives will be detected by the technology.

Precision (i.e., number of true positives in all positive alarms) is another performance measure that can be useful for understanding the validity of PLF technology. The equation for precision is presented as “Precision = true positive/true positives + false positives” and Sp is presented as “Sp = true negatives/true negatives + false positives”. While Sp will indicate the number of false positives considering true negatives (i.e., of all healthy animals or when clinical signs are absent, how many were mistakenly detected as disease animal or clinical sign), precision will indicate the number of false positives considering true positives (i.e., how many clinical signs or diseased animals were mistakenly detected by the technology). Although they are not the same, in cases where the number of true negatives does not matter (e.g., when treatment is applied to the entire group of animals regardless of whether there is a percentage of animals that are healthy or that do not present any clinical signs), precision may be used instead of Sp. This could justify why many technologies for swine or poultry did not present Sp, while for bovine all studies presented Sp. In swine production, pigs are grown in group pens and all animals are treated when there is a case of a disease [55]. However, in productions where the specific individual is treated (e.g., bovine production) Sp should be presented.

In this review, we considered that just Se and Sp, or Se and precision, could be presented by articles to support performance. However, the decision whether which performance measure is more important will vary according to the objectives of the user. If Se has little impact and Sp is very impactful in a specific situation, the user should focus on Sp instead of Se.

Analyzing performance measures can give the reader full insight into how helpful a technology will be to monitor a disease. Therefore, we find that the articles should present all relevant performance measures (Se, Sp, precision, accuracy, F1-score, PPV and PNV) to support technologies’ validity.

The nomenclature used by different studies for some performance measures may hinder interpretability. Exadaktylos et al. [34] presented a performance measure named “overall performance”, which is the same as “total accuracy” [36] and “overall accuracy” [37]. A way to overcome this would be to present the equation for the performance measure, which could help the reader to understand exactly what is being presented. However, this is not ideal since using many nomenclatures to describe the same thing could lead the reader to confusion. For example, the performance measure “Cough detection rate” is likely to be the same as Se, since many articles describe it as the ability that the technology had to detect truthful cough sounds (true positives) out of all cough sounds (true positives + false negatives). To prevent mistakenly reporting results, we decided to report the performance measure with the nomenclature used by the article.

A standard nomenclature for PLF validation studies could be a solution since every researcher would use the same terms. Although using standard terms would be ideal, we believe that it is challenging to teach and require every researcher to use the same terms. The inclusion of the confusion matrix is helpful and likely the best option as, if presented, most performance measures could be calculated (Se, Sp, accuracy, precision, F1-score, PPV, NPV). The confusion matrix is also important to assess risk of bias in the results and could be used in future meta-studies on this topic.

### 4.2. Reference Test

In this review, we decided to use “reference test” instead of “gold standard” because to our knowledge, there is no perfect test (gold standard) to identify respiratory diseases or to monitor the clinical signs. We assessed the reference test used by studies and stated what we consider to be reliable for validating PLF technologies for monitoring respiratory diseases.

The reference test is essential for supporting the performance of a technology. A technology that achieved a high performance in a study based on an unreliable reference test is likely to perform badly when applied in a commercial setting. Poorly tested technology may provide incorrect insights, consequently leading users to wrong decision-making that can negatively impact animals [56].

Vandermeulen et al. [13] and Carpentier et al. [48] used blood analysis as a reference test to confirm if the proposed technology was able to detect respiratory diseases by monitoring coughing in dairy calves. Both used the number of neutrophils as the reference test to determine if an animal was sick. Vandermeulen et al. [13] also used the Wisconsin health scoring criteria, a diagnostic tool that includes the clinical evaluation of six clinical signs caused by respiratory diseases. After comparing reference tests, blood analysis failed to correctly identify cases of bovine respiratory disease, and therefore assessing specific respiratory disease clinical signs is a better reference test to validate the technology for dairy calves [13]. It is observed in the results that the reference test affected the performance of the technology. The technology that just used blood analysis [48] as a reference test had lower Se when compared to the technology that used the Wisconsin health scoring chart [13]. If applied in field conditions, the technology with unreliable reference tests would be less effective to detect true positive cases of diseased animals.

Schaefer et al. [47] combined different clinical signs to determine if an animal was affected by respiratory disease. If the animal presented at least three clinical signs, the illness was considered as a respiratory disease. The clinical signs were high temperature, a clinical score higher than 3 (moderate to severe nasal discharge, cough, and crepitant auscultation), and low levels of white blood cells and neutrophilia. The comparison of each of these clinical signs to detect respiratory disease was presented. Fever was the most effective reference test in this study (high Se and Sp). However, caution should be taken when interpreting these results. High temperature is a clinical sign that can be caused by many diseases, so just assessing this clinical sign for exclusively detecting respiratory diseases might not be ideal, especially if applied to younger animals as these animals are susceptible to many other diseases [57]. Furthermore, the technology was tested in conditions similar to auctions and could not perform so well when applied in different environments (e.g., commercial farm).

For bovine production, there is a wider discussion on defining what could be considered a reliable reference test to detect respiratory diseases. The use of thoracic ultrasonography is considered accurate to detect respiratory diseases in bovine [58]. Studies comparing the efficiency of many clinical signs and clinical score systems to thoracic ultrasonography for diagnosing respiratory diseases were published [59,60]. Lowie et al. [60] found that spontaneous cough was the clinical sign that best indicated a calf to have respiratory diseases; however, the performance of this clinical sign is relatively poor (Se = 37.4%, Sp = 85.7%).

The use of the Wisconsin scoring system was tested and achieved a Se of 62.4% and Sp of 74.1% [61]. This result shows that the scoring chart is also not a perfect measure to be used as a reference test for detecting respiratory disease. When applying scoring systems, the criteria for diagnosing respiratory diseases in different environments should also be taken into consideration [59].

We categorized one type of reference test as audio labeled for those studies that did not assess clinical data other than cough. Overall, the methodology consists of labeling sounds from audio files that were previously recorded at animals’ sites. The labeling process is performed by a person that listens to the audio files and label cough sounds. Aerts et al. [29] compared the number of coughs that were labelled by an observer on the scene with an observer that used the audio labelled methodology previously described; this study found an underestimation of up to 94% on the number of coughs detected by the observer that labelled the audio files. Therefore, labeling sounds from recordings may not be an effective reference test. This may have artificially inflated the performance measures reported in the studies reviewed.

### 4.3. Risk of Bias

Several studies lacked key information that are considered important to assess studies’ risk of bias. Within these studies, the most common issues were related to not presenting information about the devices [32,33,43], and software [47] used. Hardware information is necessary since different sensors (i.e., different manufacturers and technology specifications) could perform differently. A full description of the software is needed to understand how the technology works and to present proof that the technology’s efficiency is supported by the software.

A few studies did not show the data (i.e., raw data or confusion matrix) that could be used to cross-reference their results [37,41,44]. Other studies lacked descriptive information of the context where the technology was applied such as the number of animals used [32] or animal production phase [37].

Descriptive information regarding animals is essential since the technology could not work as well when applied to different groups (different ages, weights, and different productions stage) or larger groups (applying the technology to a much larger group of animals). For example, for swine production most were developed for the fattening stage than the finishing stage. A technology developed and validated in a specific stage could not work well when applied in a different production stage. It is important to conduct tests in different stages to support the technology’s efficiency in different situations.

One study [38] did not provide housing information and how the devices were installed. This is necessary because technologies applied in different environments could lead to different results. Providing data on how the sensor is installed is important to understand if the technology is useful, taking in consideration its positioning; for example, a technology could work well if placed close to animals but not work when placed more distantly.

### 4.4. Can We Reliably Detect Livestock Respiratory Disease through Precision Farming?

Based on our definition of reliability, which included high technology performance achieved through a comparison against reliable reference tests, conducted in field (or similar) conditions and having low risk of bias, we identified only one study that fit these criteria [15]. This study validated a technology able to monitor pigs’ cough sounds and specific disease detection (Mycoplasma hyopneumoniae, porcine reproductive and respiratory disease, and postweaning multisystemic wasting syndrome) in field conditions. The technology is described as a low-cost solution that is suitable for smaller farms. All information to assess risk of bias is described by the study. Two reference tests were used: labeling cough sounds from video and audio recordings (video was used to ensure that a sound was correctly labeled), and blood analysis to determine the type of respiratory disease affecting the pigs. The cough sound was linked to the disease affecting the pigs to validate the algorithm for specific disease detection. The technology achieved high performance for both cough sound detection and specific disease detection. The only limitation of the study is the low sample size.

Other two studies complied with most of our criteria but had a high risk of bias [37,47]. One swine study [37] did not describe the study population and the raw data supporting their performance results. The other study monitored respiratory diseases with infrared thermography in bovine production [47]; however, it did not provide full software description.

A limitation of our approach may have been that setting the performance threshold at >90% resulted in leaving several relevant articles out of further analysis. Another limitation is that we have not assessed all the PLF proceedings from conferences and companies’ internal validations available online.

## 5. Conclusions

We evaluated PLF technologies to monitor respiratory diseases in swine, bovine, and poultry productions. One study met all the criteria set in this review to be considered reliable. Some studies achieved high performance, but it is unclear if the technologies would perform well when applied in real-life situations. Many studies were only conducted on laboratory conditions or used an unreliable reference test. We identified issues with how studies report their validation tests, such as not reporting all relevant performance measures, low performance measures, different nomenclature for the same performance measure within studies, and not reporting all necessary information to assess risk of bias.

We encourage future studies to improve how the methodology and results are reported so that all important information is provided for readers to have a full understanding on the effectiveness of a technology.

The fact that some studies have had good performances indicate that, in the future, it might be possible to rely on PLF technologies for automatic monitoring of respiratory diseases.

## Figures and Tables

**Figure 1 animals-13-01273-f001:**
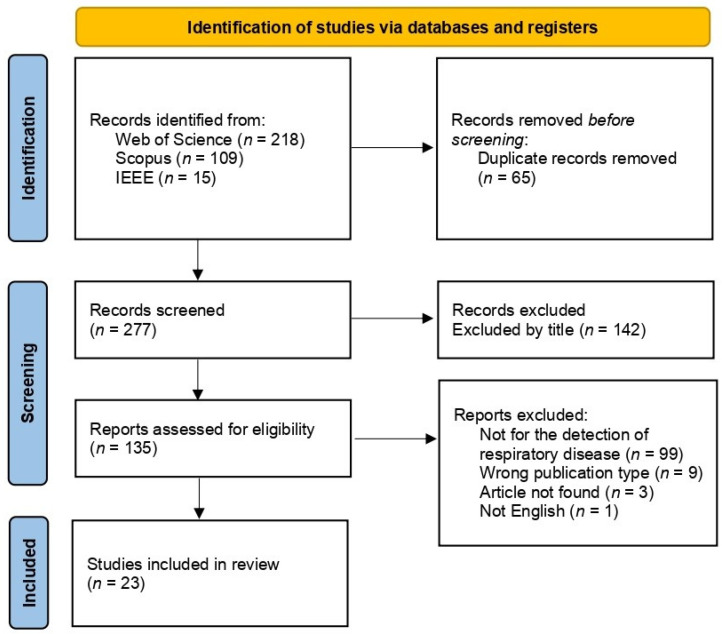
Modified Preferred Reporting Items for Systematic Reviews and Meta-Analyses (PRISMA) flow diagram [24] containing information of the strategy for the systematic review search and study selection. The PRISMA checklist is presented as Appendix A.

**Table 1 animals-13-01273-t001:** Search strings for the literature search.

Species Terms	Technology Terms	Type of Conditions Terms
Dairy cow(s)	Precision Livestock Farming	Respiratory Disease(s)
Cow	Noninvasive Technology	Cough
Cattle	Smart sensor	Fever
Calf	Smart Farming	BRD
Calves	Automated technology	Vocalization
Pig	Online health monitoring	Infectious disease(s)
Sow	Computer vision	Sneeze
Swine	Cough recognition	Respiratory disease detection
Broiler	Sound analysis	
Laying hen	Sound classification	
Chicken	Convolutional neural network	
Poultry		
Goat		
Sheep		
Ewe		
Lamb		

**Table 2 animals-13-01273-t002:** Information about sensor type, performance measures, study conditions, and reference test of included articles for swine, poultry, and bovine productions.

Species	Study	Sensor Type	Performance Measures	Study Conditions	Reference Test
Swine	[30]	Sound Based	Positive cough recognition	Laboratory	Remote audio labeling
	[31]	Sound Based	Positive cough recognition	Laboratory	Remote audio labeling
	[32]	Sound Based	Accuracy	Field	Live audio labeling
	[33]	Sound Based	Accuracy	Field	Live audio labeling
	[34]	Sound Based	Correct identification ratio	Laboratory	Remote audio labeling
	[35]	Sound Based	Correct identification ratio	Field	Remote audio labeling
	[36]	Sound Based	Accuracy	Field	Live audio labeling
	[37]	Sound Based	Sensitivity, Precision, Accuracy, and cough detection rate	Field	Remote audio labeling and blood analysis
	[15]	Sound Based	Sensitivity, Precision, cough detection rate, and F1-score	Field	Video labeling and blood analysis
	[38]	Sound Based	Word error rate	Laboratory	Remote audio labeling
	[39]	Sound Based	Sensitivity, Specificity, Precision, Accuracy, and F1-score	Field	Remote audio labeling
	[40]	Sound Based	Sensitivity, Specificity, Precision, Accuracy, and F1-score	Field	Remote audio labeling
	[41]	Sound Based	Sensitivity, Precision, Accuracy, and F1-score	Field	Remote audio labeling
Poultry	[42]	Sound Based	Sensitivity, Specificity, and Accuracy	Laboratory	PCR
	[43]	Sound Based	Sensitivity, Precision, and Accuracy	Laboratory	Remote audio labeling
	[14]	Sound Based	Sensitivity, Specificity, and Precision	Laboratory	Remote audio labeling
	[44]	Sound Based	Accuracy	Laboratory	PCR
	[45]	Sound Based	Sensitivity, Precision, Accuracy, and F1-score	Field	Remote audio labeling
	[46]	Sound Based	Sensitivity, Precision, Accuracy, and F1-score	Laboratory	Video labeling and PCR
Bovine	[47]	Image	Sensitivity, Specificity, PPV, NPV, and Cut off value	Field	Clinical assessment
	[13]	Sound Based	Sensitivity, Specificity, and Precision	Field	Clinical assessment and blood analysis
	[48]	Sound Based	Sensitivity, Specificity, and Precision	Field	Blood analysis
	[16]	Accelerometer	Sensitivity, Specificity, Accuracy, and MCC	Field	Clinical assessment

**Table 3 animals-13-01273-t003:** Evaluation of the descriptions of eight essential pieces of information in the studies to assess risk of bias.

Species	Study	Study Conditions	Housing	Hardware	How It Was Installed	Software	Population Description	Number of Animals	Raw Data	Risk of Bias
Swine	[30]	✔	✔	✔	✔	✔	✔	✔	✔	low
	[31]	✔	✔	✔	✔	✔	✔	✔	✔	low
	[32]	✔	✔	✖	✔	✔	✔	✖	✔	high
	[33]	✔	✔	✖	✔	✔	✔	✔	✔	high
	[34]	✔	✔	✔	✔	✔	✔	✔	✔	low
	[35]	✔	✔	✔	✔	✔	✔	✔	✔	low
	[36]	✔	✔	✔	✔	✔	✔	✔	✔	low
	[37]	✔	✔	✔	✔	✔	✖	✔	✖	high
	[15]	✔	✔	✔	✔	✔	✔	✔	✔	low
	[38]	✔	✖	✔	✖	✔	✖	✔	✔	high
	[39]	✔	✔	✔	✔	✔	✔	✔	✔	low
	[40]	✔	✔	✔	✔	✔	✔	✔	✔	low
	[41]	✔	✔	✔	✔	✔	✔	✔	✖	high
Poultry	[42]	✔	✔	✔	✔	✔	✔	✔	✔	low
	[43]	✔	✔	✖	✔	✔	✔	✔	✔	high
	[14]	✔	✔	✔	✔	✔	✔	✔	✔	low
	[44]	✔	✔	✔	✔	✔	✔	✔	✖	high
	[45]	✔	✔	✔	✔	✔	✔	✔	✔	low
	[46]	✔	✔	✔	✔	✔	✔	✔	✔	low
Bovine	[47]	✔	✔	✔	✔	✖	✔	✔	✔	high
	[13]	✔	✔	✔	✔	✔	✔	✔	✔	low
	[48]	✔	✔	✔	✔	✔	✔	✔	✔	low
	[16]	✔	✔	✔	✔	✔	✔	✔	✔	low

✔—Information provided in the article. ✖—Information not provided in the article.

**Table 4 animals-13-01273-t004:** PLF performance measures for poultry production.

	Sensitivity (%)	Specificity (%)	Precision (%)	Accuracy (%)	F1-Score (%)
[42]	93.30	96.73	N/A	91.15	N/A
[43]	85.20	N/A	86.60	97.60	N/A
[14]	66.70	N/A	88.40	N/A	N/A
[44]	N/A	N/A	N/A	97.00	N/A
[45]	94.10	N/A	94.40	93.80	94.20
[46]	96.60	N/A	96.54	98.50	97.33

N/A—Not Available.

**Table 5 animals-13-01273-t005:** PLF performance measures for bovine production.

	Sensitivity (%)	Specificity (%)	Precision (%)	Accuracy (%)	PPV (%)	NPV (%)
[47]	100.00	97.40	N/A	N/A	86.30	100.00
[13]	50.30	99.20	87.50	N/A	N/A	N/A
[48]	41.40	99.90	94.20	N/A	N/A	N/A
[16]	54.00	95.00	N/A	75.00	N/A	N/A

N/A—Not available.

**Table 6 animals-13-01273-t006:** PLF performance measures for swine production.

	Sensitivity (%)	Specificity (%)	Precision (%)	Accuracy (%)	F1-Score (%)	Cough Detection Rate (%)
[30]	N/A	N/A	N/A	N/A	N/A	94.80
[31]	N/A	N/A	N/A	N/A	N/A	94.80
[32]	N/A	N/A	N/A	N/A	N/A	90.00
[33]	N/A	N/A	N/A	86.20	N/A	N/A
[34]	N/A	N/A	N/A	N/A	N/A	82.00
[35]	N/A	N/A	N/A	N/A	N/A	88.00
[36]	N/A	N/A	N/A	86.20	N/A	85.50
[37]	92.00 ^1^	N/A	90.80 ^1^	91.00 ^1^	N/A	94.00
[15]	98.60 ^1^	N/A	95.50 ^1^	N/A	94.70 ^1^	99.00
[39]	97.72	95.01	96.81	96.68	97.26	97.72
[40]	96.80	93.20	95.50	95.40	96.20	96.80
[41]	96.51	N/A	98.41	97.35	97.46	96.51

^1^ Performance measures of specific disease detection. N/A—Not available.

## Data Availability

No new data were created or analyzed in this study. Data sharing is not applicable to this article.

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
