# Peer review of "Can We Reliably Detect Respiratory Diseases through Precision Farming? A Systematic Review"

_animals, 2023, doi:10.3390/ani13071273_

Round 1

Reviewer 1 Report

The review paper gives a good overview and summary of the scientific papers in the domain of PLF/respiratory health detection.

Suggestions for improvement are:

Line 13: sentence should be reformulated

Rules 14 and 20: Give a clear definition of “monitoring respiratory diseases” Do you main symptoms (coughs, sneezes, rales, etc.) or the type of disease/pathogen ?

Line 22: define “valid reference test”

Line 29: Validated & patented commercial PLF solutions for monitoring swine respiratory health already exist, e.g. https://bi-animalhealth.com/swine/ihm/soundtalks

Line 60: remove “Norton and Berckmans”

Line 63: True/False positives and negatives (based on event classification) are not always linked to disease status. Other factors (e.g. air quality) can also cause coughs or sneezes.

Line 95: Proceedings from relevant PLF conferences , such as ECPLF should be added to the search list.

Line 100: replace ‘precision livestock farming’ by ‘Technology terms”.

Line 153: In many countries fattening stage is used for pigs between 30 kg until slaughter

Line 201: the definition of misclassifications seems to conflict with the definition in 3.1 (line 227)

Lines 263+ Discussion This part should be integrated in part 3 (Results) because there is a lot of overlap

Line 554+: Conclusions: A summary must be added with  the most relevant quantitative results of the study.

Author Response

Dear reviewer, please find attached our rebuttal letter to all comments in our manuscript. Thank you for taking the time to evaluate our work.

Reviewer 2 Report

Dear Authors, this review is interesting but there are some concerns that need improvement. In particular, the discussion should be improved (see last comment below) because the benefit of this study does not result clear. In addition, some aspects mentioned in materials and methods, results and discussion look repetitive and they could be better reported.

Lines 12-16: please rephrase, unclear sentences.

Line 60: fix the reference

I do not understand why you dedicate a lot in the text to specify the relevance of selecting papers in which sensitivity, specificity, accuracy or precision are present, whilst in most articles of Table 6 this information is not completely available.

On which basis did you select >90% Se and Sp to include the literature study in your study?

Lines 381-385: could it be that the reference tests affected the results of some studies that you did not select based on the low Se and Sp?

Line 424: please rephrase.

Lines 549-550: is the 3+1 studies referring to the total of 6 studies mentioned few lines before, right? What about the other 2 studies? (they should be 6 in total)

Line 551: is it 3 studies out of 6? This part of the text should be better written. What are the main “drawbacks” of the other studies that you selected? You mentioned that from the initial selection you then kept 23 studies but in the end only 6 (that then become 1), seem to be well done. You should explain this better.

Finally, since only one study effectively outperformed the others in respect to the required fields, could you summarize their main findings and specificities that characterize this study? Otherwise it is not clear what is the benefit of this literature review. What is the main finding of this study? How the results of this study bring improved knowledge? What should authors do to improve the quality of their studies? All these recommendations could be a valid support to better conclude your study.

Author Response

Dear reviewer, thank you for evaluating our work. Please find attached a rebuttal letter to all comments in our manuscript.

Round 2

Reviewer 2 Report

Dear Authors,

thank you for the improvements introduced in the text. I now believe it is much clearer and suitable for publication.